# Learning A Low-Level Vision Generalist via Visual Task Prompt

## ABSTRACT

Building a unified model for general low-level vision tasks has important research and practical value. However, existing methods still face challenges when dealing with diverse low-level vision problems. Multi-task restoration approaches can simultaneously address various degradation-to-clean restoration tasks, while their applicability to tasks with different target domains (e.g., image stylization) remains limited. Existing methods like PromptGIP that can handle tasks with multiple input-target domains mainly rely on the Masked Autoencoder (MAE) training paradigm. Unfortunately, these approaches are restricted by coupling to the ViT architecture, resulting in suboptimal image reconstruction quality. In addition, they tend to be sensitive to prompt content and often fail when handling more tasks that involve low-frequency information processing, such as color and style. In this paper, we present a Visual task Prompt-based Image Processing (VPIP) framework to address the above challenges. This framework employs the visual task prompt to process tasks with different input-target domains. Besides, it provides the flexibility to select a backbone network suitable for various low-level vision tasks. A prompt cross-attention mechanism is introduced to deal with the information interaction between the input and prompt information. Based on the VPIP framework, we train a low-level vision generalist model, namely GenLV, on 30 diverse tasks. Experimental results show that GenLV can successfully address a variety of low-level tasks, and it significantly outperforms existing methods both quantitatively and qualitatively.

## CCS CONCEPTS

• **Computing methodologies** → **Image representations**; **Reconstruction**; **Computational photography**.

## KEYWORDS

General Low-Level Vision, Image Restoration and Enhancement, Multi-task Learning, Visual Prompt

## 1 INTRODUCTION

low-level vision comprises a multitude of tasks that manipulate and enhance the pixel-level information of images. These tasks include but are not limited to image restoration, image enhancement, image feature extraction and image stylization. Over the years, a flurry of methods have been proposed to address various low-level vision tasks, many of which have achieved commendable performance for specific individual tasks [6, 11, 48]. However,

Permission to make digital or hard copies of all or part of this work for personal or classroom use is granted without fee provided that copies are not made or distributed for profit or commercial advantage and that copies bear this notice and the full citation on the first page. Copyrights for components of this work owned by others than the author(s) must be honored. Abstracting with credit is permitted. To copy otherwise, or republish, to post on servers or to redistribute to lists, requires prior specific permission and/or a fee. Request permissions from permissions@acm.org.

*ACM MM, 2024, Melbourne, Australia*

© 2024 Copyright held by the owner/author(s). Publication rights licensed to ACM.
ACM ISBN 978-x-xxxx-xxxx-x/YY/MM
https://doi.org/10.1145/nnnnnnn.nnnnnnn

the custom development of task-specific models often proves to be time-consuming and labor-intensive. In recent years, there has been a significant tendency in artificial intelligence technology towards the development of general models. In the realm of Natural Language Processing (NLP), Large Language Models (LLMs) such as the GPT-series [4, 33] have exhibited remarkable performance. Analogous research has also incrementally emerged in the field of computer vision, exemplified by models like Segment Anything Model (SAM) [24] and Track Anything Model (TAM) [45]. However, these explorations have mainly focused on perceptual high-level vision tasks, while research concerning general models for low-level vision tasks remains notably insufficient.

Designing a general model for low-level vision presents significant challenges in several aspects. Firstly, due to the diversity of low-level vision tasks, different types of tasks may have distinct input and target domains (e.g., image restoration and stylization). Therefore, unifying a wide range of low-level vision tasks within a single model framework is challenging. Existing models that solve multiple low-level vision tasks are often designed to deal with a specific task category. For instance, AirNet [26] and PromptIR [35] focus on simultaneously processing several restoration tasks, i.e., restoring degraded images with several specific degradation to clean ones. However, these models are not capable to jointly process image feature extraction or stylization tasks that accept clean images as input. Secondly, for low-level vision tasks, the pixel-level image reconstruction and generation quality is the most important factor in evaluating the model effectiveness. However, existing general vision models frequently emphasize perceptual accuracy while neglecting the model's image reconstruction capability. Consequently, when tackling low-level vision tasks, the generated results frequently suffer from unsatisfactory image quality. For example, MAE-VQGAN [3] utilizes discrete feature representations for image reconstruction, often resulting in unacceptable structural differences between the reconstructed results and the input images [30]. Painter [40] and PromptGIP [30] employs ViT-based backbone network [17]. As a result, their results often lack fine details and occasionally exhibit blocking artifacts. Furthermore, solving a broader range of low-level vision tasks involves processing high- and low-frequency information simultaneously, which also poses a significant challenge to method design. PromptGIP [30] performs well within a specific task range, but may fail when the number of tasks increases, especially when more low-frequency information processing (e.g., color and style) is involved. The Painter [40] model trained under the same task settings exhibits similar or even more severe issues, as shown in Figure 5. We attribute these phenomena to the Masked Autoencoder (MAE)-based training paradigm, which makes the models sensitive to the content, especially low-frequency information, of the visual prompt.

To address the above issues of constructing a general low-level vision model, we propose a new Visual task Prompt-based Image Processing (VPIP) framework. This framework comprises three key components: an end-to-end image processing main network,

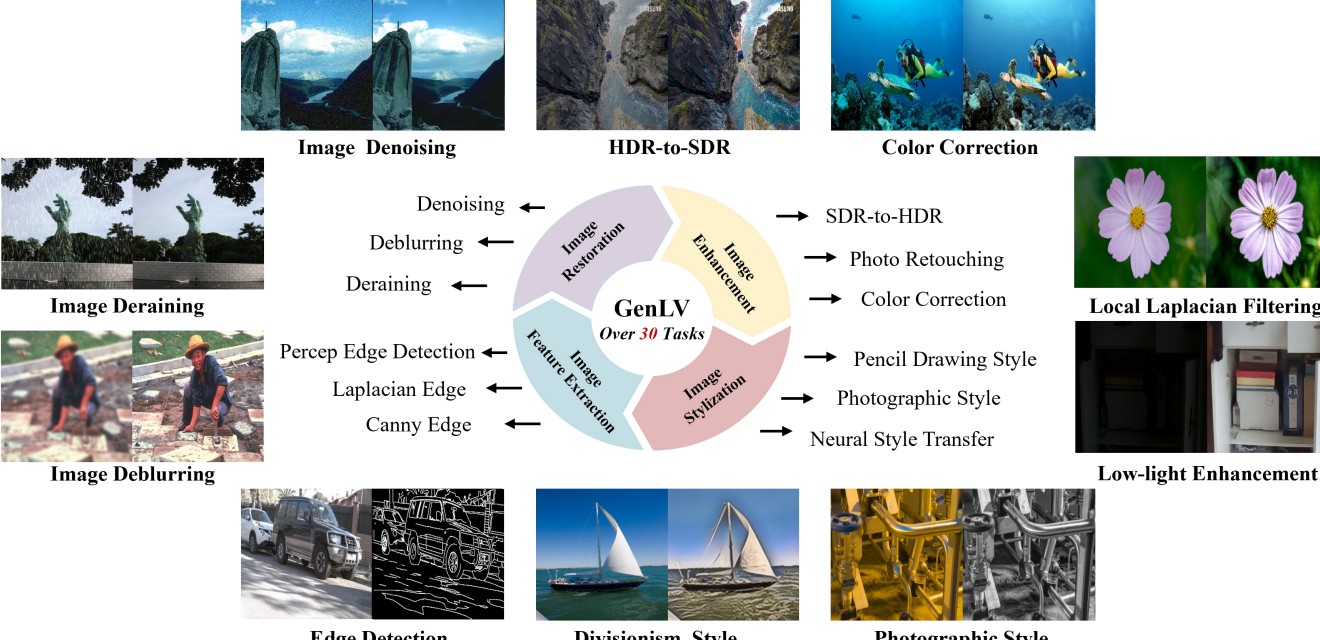

**Figure 1: Our proposed low-level vision generalist model, GenLV, can handle diverse tasks with various input/target domains.**

an encoder sub-network for processing the visual prompt, and an information interaction mechanism that introduces prompt information into the main network to guide task-specific processing. For the specific model design, we first employ a powerful backbone network, X-Restormer [9], which is designed for general image restoration tasks, as the main network. Similar to PromptGIP and Painter, we represent different low-level vision tasks as input-target image pairs, serving as the visual task prompt. The prompt encoder sub-network processes the task prompt into latent representations. Inspired by text-to-image models, such as the Stable Diffusion model [38], which incorporate the text prompt into the denoising process, we introduce visual task prompt representations into the main network using a cross-attention mechanism.

The advantages of our proposed VPIP framework are manifold: 1) By utilizing the visual task prompt, our framework effectively solve the problem of varying input-target domains across different low-level vision tasks. 2) Not restricted to the MAE training paradigm, our approach mitigates the impact of prompt content on the task representation, enhancing the robustness of the task prompt. 3) Our framework allows flexibility in selecting backbone networks suitable for low-level vision tasks, thereby improving image reconstruction quality. 4) Instead of computing global attention for grid-like representations consisting of four image features of PromptGIP, our approach, based on cross-attention calculation, effectively reduces the attention computation cost.

To evaluate the effectiveness of our method, we construct 30 diverse tasks to train the model. The trained low-level vision generalist model, namely GenLV, can successfully process the various tasks with different input and target domains, as show in Figure 1. We also conduct comprehensive experiments to compare our method and existing approaches in Section 4. Extensive results demonstrate that GenLV significantly outperforms the other methods.

## 2 RELATED WORK

**Low-Level Vision.** Over the past decade, the field of low-level vision has undergone significant advancements, largely attributable to the integration of deep learning techniques. Classic low-level vision tasks include but not limited four categories: image restoration, image enhancement, image feature extraction and stylization. Image restoration aims to restore the high-quality clean image from observations degraded by a variety of factors, including low-resolution [16], noise [50], blur [1], JPEG compression [15] and bad weather, such rain [46] and haze [47]. Image enhancement [36] is concerned with the alteration of specific image attributes, such as color [19], sharpness [2], exposure [10] and brightness [8], in order to make the image more suitable for a particular task or viewer. Image feature extraction, like edge detection [7], focuses on extracting the low-level features from the image, which can help downstream enhancement and understanding tasks. Image stylization is to create visually appealing images with a specific style or aesthetic [23]. Despite the advancements, current methods often rely on specialized datasets and customized network architectures, which limits the practical application of these methods.

**Prompt Learning.** In the NLP field, the concept of prompting is initially to supply manually selected in-context information to a pretrained model for implementing the target task [4]. Instead of using manual prompt, many follow-up works propose to treat the prompt as task-specific vectors to adapt model for various tasks [20, 25]. Prompt learning techniques have also been applied in the field of computer vision, where they have proven effective in modeling task-specific instructions across various applications [21, 51]. Notably, MAE-VQGAN [3] and Painter [40] leverage the flexibility of prompting to unify a variety of vision tasks. These models demonstrate impressive performance on high-level tasks, such as semantic

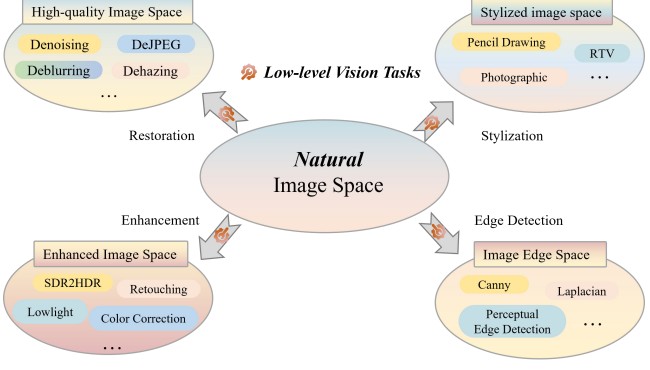

**Figure 2: Diverse low-level vision tasks. Different categories of tasks differ in terms of target domains. It presents a significant challenge to build a low-level vision generalist model.**

segmentation, by constructing the grid-like prompt. However, their effectiveness in low-level vision tasks is insufficient [30]. In the realm of low-level vision, PromptGIP [30] proposes an MAE-based framework and use the grid-like visual prompt to handle 15 cross-domain image processing tasks. Despite this, as the number and complexity of tasks increases, the effectiveness of this method decreases significantly. Besides, the training paradigm employed by PromptGIP is deeply intertwined with the ViT architecture, which largely limits the image reconstruction quality of its output.

**Multi-task Image Restoration.** Multi-task image restoration aims to train a single model to handle multiple restoration tasks simultaneously. Existing multi-task image restoration methods can be categorized into two groups. The first group of methods aim to process real-world images with unknown degradation, emphasizing the modeling of complex real-world degradation. The representative approaches include BSRGAN [49] and Real-ESRGAN [41]). In contrast, the second group of methods like DASR [39] and Air-Net [26] are developed to handle several specific restoration tasks with predefined degradation. These methods mainly focus on designing better modules for multi-task learning to maximize network capability of task-specific restoration performance. Some current works such as ProRes [32] and PromptIR [35] are proposed to leverage a learnable prompt from the input image for better multi-task restoration. However, all these approaches are limited to solving the degradation-to-clean restoration problem, and lack the ability to deal with a broad range of cross-domain low-level vision tasks. Unlike these approaches, our method aims to construct a low-level vision generalist model, which is not only capable of image restoration, but also excels at handling a wider range of cross-domain tasks, including enhancement, feature detection and stylization.

## 3 APPROACH

### 3.1 Representative Low-Level Vision Tasks

Low-level vision tasks encompass a range of pixel-level manipulations, including image restoration, enhancement, feature extraction, stylization, etc. Each task uniquely transforms an input image space to a specific target domain. For example, the target domain of image restoration is high-quality (HQ) image space $\Omega_{HQ}$, while the

outputs of edge detection are edges maps $\Omega_{Edge}$. Formally, given an arbitrary input image $I$, the low-level vision task can be defined as $\mathcal{T}_{task} : \Omega_S \rightarrow \Omega_T$, where $\Omega_S$ and $\Omega_T$ denote the source image space and the target image space, respectively. According to the target domain, low-level vision tasks generally fall into the following categories as:

$$
\begin{aligned}
\text{Restoration:} \quad & \mathcal{T}_{Res} : \Omega_S \rightarrow \Omega_{HQ}, \\
\text{Enhancement:} \quad & \mathcal{T}_{Enh} : \Omega_S \rightarrow \Omega_{Enh}, \\
\text{Edge Detection:} \quad & \mathcal{T}_{Edg} : \Omega_S \rightarrow \Omega_{Edg}, \\
\text{stylization:} \quad & \mathcal{T}_{Sty} : \Omega_S \rightarrow \Omega_{Sty}.
\end{aligned}
\tag{1}
$$

Each category encompasses a variety of tasks, as presented in Figure 2. Our goal is to address all these tasks through a unified model.

### 3.2 Problem Formulation

Existing low-level vision methods are typically designed for specific tasks, which inherently restricts their applicability to tasks with different target domains. Taking the image restoration model as an example, they accept low-quality images as input and predict the high-quality output as:

$$
I_{out} = \mathcal{F}_{\mathcal{T}_{Res}}(I_{in}; \Theta) \in \Omega_{HQ}, \tag{2}
$$

where $\mathcal{F}_{\mathcal{T}_{Res}}$ represents the restoration model parameterized by $\Theta$. The restoration model can accommodate various restoration tasks through incorporating multiple degradations into training, such as blur, denoising and deraining, given that the output image space of these tasks are the same, i.e., $\Omega_{HQ}$. However, this kind of model cannot be extended to simultaneously implement tasks like edge detection, which targets a completely different output modality.

To train a low-level vision generalist model that can process cross-domain tasks, our approach employs a unified framework capable of handling various cross-domain tasks by utilizing additional image pair as the prompt $[P_{\Omega_S}, P_{\Omega_T}]$. It can be denoted as:

$$
I_{out} = \mathcal{F}_{\mathcal{T}}(I_{in}, [P_{\Omega_S}, P_{\Omega_T}]; \Theta). \tag{3}
$$

This formulation allows diverse task mappings to be represented by intuitive image pairs, which marks a significant difference from conventional low-level vision models, offering a more holistic and adaptable approach to broad cross-domain low-level vision tasks.

### 3.3 Low-Level Vision Generalist Model

In this section, we illustrate the specific design of our low-level vision generalist model, as shown in Figure 3. The overall approach is predicated on the Visual task Prompt-based Image Processing (VPIP) framework. A powerful image processing network and a prompt encoder network are used to process the input image and the prompt images. A new prompt cross-attention mechanism is introduced to achieve the information interaction among latent representations of the input image and prompt images.

**VPIP Framework** consists of an end-to-end image processing main network, a prompt encoder network and a prompt interaction mechanism. Given an input image $I_{in}$, it is initially processed to a high-dimensional latent feature $z_{in}$ through the encoder. In parallel, the paired prompt images $[P_{\Omega_S}, P_{\Omega_T}]$ are fed into the prompt encoder to generate two high-dimensional representations $[z^P_{\Omega_S}, z^P_{\Omega_T}]$, both of which with the same spatial size as $z_{in}$. Following this, the

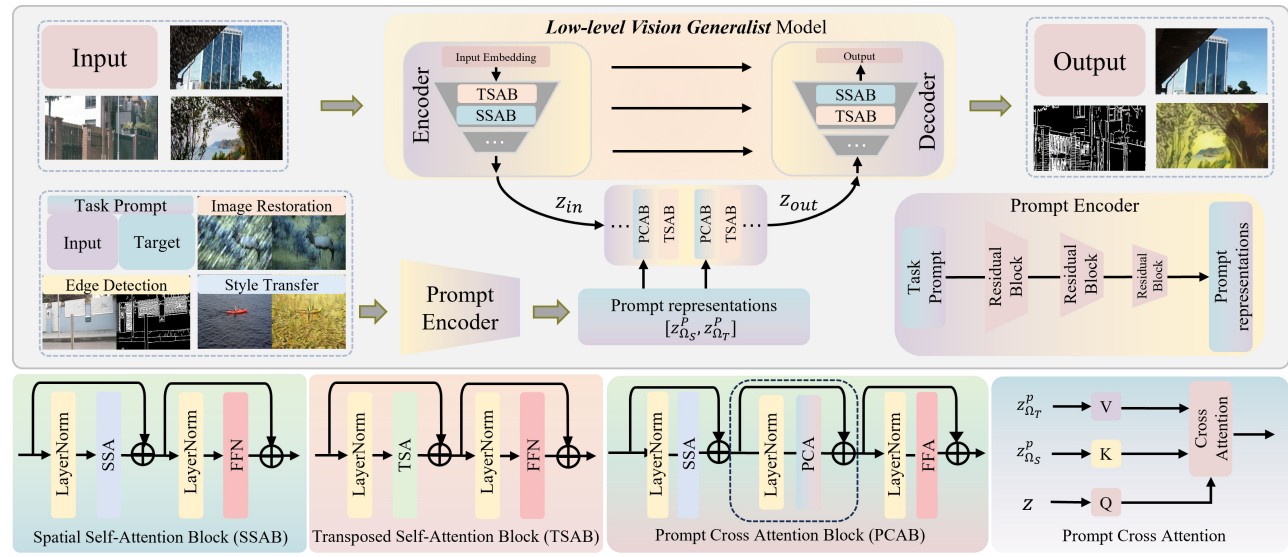

Figure 3: Overall approach of our low-level vision generalist model, GenLV.

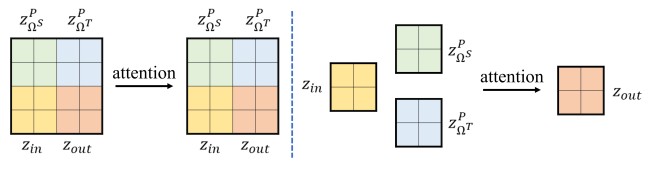

(a) Global self-attention      (b) Prompt cross-attention

Figure 4: Comparison of two attention mechanisms.

information interaction is implemented between $z_{in}$ and the pair $[z^P_{\Omega_S}, z^P_{\Omega_T}]$ and results in the processed latent representation $z_{out}$. The final step is to reconstruct the output image from $z_{out}$ via the decoder in the main network. Unlike previous approaches such as Painter and PromptGIP, which rely on MAE-based framework and require binding with the ViT architecture, our VPIP framework allows flexible selection of backbone networks suitable for low-level vision tasks as the image processing main network.

**Image Processing Backbone** plays a crucial role in the image reconstruction quality for low-level vision tasks. Since different low-level vision tasks often have different requirements for network capability, the most important criterion for selecting a backbone network is its task generality. Due to the lack of work investigating task generality across a wide range of low-level tasks, we focus more on the model performance for image restoration. A recent work conduct a detailed study on the backbone network for restoration tasks and propose a general backbone network, X-Restormer [9], suitable for multiple various restoration tasks. Therefore, we simply adopt similar architecture as our main network[1]. Specifically, the main network employs a U-shape architecture, where downsampling and upsampling operations are performed three times,

---

[1]We also conduct an extensive study on the model performance of different backbone network, and the detailed results are presented in the Supplementary Material.

and skip connections are added from the encoder to the decoder at the same scale. The basic modules to construct the network are Transposed Self-Attention Block (TSAB) and Spatial Self-Attention Block (SSAB), which deal with the channel-wise global information interaction and the spatial information interaction, respectively. The structures of TSAB and SSAB are shown in Figure 3. The implementation details of their attention mechanism can be referred to the OCA in HAT [12] and the MDTA in Restormer [48].

**Prompt Cross-attention** is designed to perform information interaction among the prompt and input representations. Prompt-GIP [30] demonstrates that calculating global attention in the feature space can effectively incorporate task prompt information into image processing. However, this approach is tightly coupled with the ViT architecture and is relatively inefficient in terms of the attention computation. Inspired by the Stable Diffusion [38] model, which utilizes cross-attention to apply text prompt to the denoising UNet, we adopt a similar mechanism to introduce visual prompt information into the image processing network. As depicted in Figure 3, the Prompt Cross-Attention Block (PCAB) is implemented by adding a PCA module to the standard SSAB and is integrated at the bottom of the U architecture. To calculate the PCA, the *query* ($Q$), *key* ($K$) and *value* ($V$) are first generated by $1 \times 1$ convolutions from the input representation $z_{in}$, prompt input embedding $z^P_{\Omega_S}$ and prompt target embedding $z^P_{\Omega_T}$. Then, the standard attention is computed to obtain the output representation $z_{out}$. Compared to calculating global self-attention on the grid-like features consisting of four image representations across the entire network, our prompt cross-attention calculated based on the size of one image representation in just a few blocks (i.e., PCAB) is much more efficient in terms of attention computation, as shown in Figure 4.

**Prompt Encoder** is employed to encode the prompt images into deep representations that can be used for information interaction. We simply utilize a series of standard residual blocks spaced by multiple downsampling operations to build the encoder network.

# 4 EXPERIMENTS AND ANALYSIS

## 4.1 Experimental Setup

**Task Settings.** We train the models on **30** diverse low-level tasks, as follows: 1) **Image Restoration:** Following PromptGIP [30], ten classic degradation types are considered including Gaussian noise, Gaussian blur, Poisson noise, salt & pepper noise, JPEG compression, ringing artifacts, R-L algorithm [37], inpainting, haze and rain. During the training, the on-the-fly data pairs are generated on ImageNet [14] for the first eight types. Simple mixed degradations are also considered for training. ITS dataset [27] and Rain13K [22] are utilized for dehazing and deraining. A simple additive rain model is also employed for synthetic data. For testing, a mixed dataset, Common528 [30], composed of several low-level vision benchmark datasets is employed. 2) **Image Enhancement:** This category includes eight tasks: low-light enhancement (LLE), photo retouching, local Laplacian filtering [2] (LLF), multi-scale tone manipulation (MTM) [18], underwater image contrast enhancement (ICE) based on histogram equalization, underwater image color correction (ICC) based on the DIVE+ software, image SDR-to-HDR and HDR-to-SDR [13]. The LOL dataset [42] is used for LLE, and expert-C retouched images of the Adobe-MIT Fivek dataset [5] are used for retouching, LLF, and MTM. The UIEB dataset [28] is utilized for the two underwater image enhancement tasks. 3) **Image Edge Detection:** This category includes three edge detection tasks: Canny operator, Laplacian operator and a perceptual edge detection (PED) [34]. 4) **Image Stylization:** Nine style are chosen, including pencil drawing [31], photographic style [2], relative total variation (RTV) [44], Vermeer style, JOJO style, Raphael style, Fauvism style, Divisionism style and Cloisonnism style. Expert-C retouched images of Adobe-MIT Fivek dataset [5] are also used to generate the image pairs. Data of the first three styles are implemented via available toolkit, and the last six neural styles are generated by a state-of-the-art style transfer method AdaAttN [29].

**Implement details.** For the backbone network, we adopt the similar setting as the original X-Restormer [9]. From level-1 to level-4, the numbers of consecutive blocks (each block contains a TSAB and an SSAB) are [2, 4, 4, 4], attention heads in TSA and SSA are both [1, 2, 4, 8], and channel numbers are [48, 96, 192, 384]. For the prompt encoder network, we employ four residual blocks for each downsampling level. During the training, the input size is set to $256 \times 256$ for the input image and prompt images. $L_1$ loss is utilized as the loss function. AdamW optimizer with $\beta_1 = 0.9$ and $\beta_2 = 0.99$ is adopted with an initial learning rate of $1e^{-4}$. The batch size is set to 64 and total training epochs are 30.

## 4.2 Quantitative Results

The quantitative results for various low-level vision tasks are presented in Table 1, Table 2 and Table 3. Given that not all existing methods are capable to handle tasks across different target domains, our primary comparative experiments are centered on restoration tasks in Table 1. We consider three distinct experimental settings. The first setting utilizes a pretrained model, i.e., Real-ESRGAN [41], which is capable of handling a variety of restoration tasks. The second setting is based on the training configuration outlined in our paper, but it solely focuses on image restoration tasks. The third setting involves training on the all 30 low-level vision tasks.

**Ablation Study on Visual Prompt.** Since the models without using task prompt cannot process tasks with different target domains, we conduct the ablation study of the visual task prompt on restoration tasks. In Table 1, ViT⋆ and X-Restormer⋆ are two end-to-end models only trained on image restoration tasks, while ViT-VPIP⋆ and GenLV⋆ (the GenLV model can also be represented as X-Restormer-VPIP) are models based on our VPIP framework, utilizing ViT and X-Restormer as their backbone respectively. Upon the incorporation of prompt learning, both ViT-VPIP⋆ and GenLV⋆ exhibit substantial performance gains over ViT⋆ and X-Restormer⋆ in most restoration tasks. This demonstrates the effectiveness of the visual prompt in facilitating the backbone network to better handle various tasks. It is noteworthy that X-Restormer⋆, without using visual prompt, struggles with the dehazing task, achieving only 16.73dB. A similar phenomenon also occurs for the multi-task restoration method PromptIR [35]. In contrast, GenLV⋆ tackles it considerably better, reaching 25.63dB. All these results show the effectiveness of our proposed VPIP framework.

**Influence of Backbone Network.** In Table 1, when trained on the same setting, the performance of models using X-Restormer as the backbone network (i.e., X-Restormer⋆, GenLV⋆ and GenLV†) significantly surpasses that of models using ViT (i.e., ViT⋆, ViT-VPIP⋆ and ViT-VPIP⋆). This observation suggests that an appropriate backbone network is important for low-level vision tasks generalist models, and ViT architecture may limit the model performance.

**Comparison with other methods.** In Table 1, GenLV⋆ outperforms the state-of-the-art blind SR method Real-ESRGAN [41] and multi-task restoration method PromptIR [35], when only considering image restoration tasks. Note that we retrain the PromptIR model on the same setting for fair comparison (the original PromptIR is trained only on 4 tasks). By employing the ViT network, PromptGIP⋆ trained on restoration tasks performs better than ViT-VPIP⋆, due to more attention computation. However, as more tasks are involved, ViT-VPIP† outperforms PromptGIP† and Painter† instead, showing the superiority of our framework for solving more diverse tasks. In Table 2 and Table 3, we further show the quantitative results on broader low-level vision tasks. Only methods capable of solving tasks across different target domains are considered in the comparison. The models employed VPIP framework outperform Painter and PromptGIP on a variety of low-level vision tasks.

## 4.3 Visual Results

In Figure 5, we present the visual comparison of our GenLV with Painter and PromptGIP across various low-level vision tasks. From a holistic perspective, GenLV produces results that are more consistent with the ground truth, especially in aspects such as color and brightness. In contrast, the results produced by Painter and PromptGIP are easily affected by errors in low-frequency information, manifesting as color anomalies or even incorrect task execution. Rather that our method where prompt information can accurately serve as task instruction, Painter and PromptGIP appear to be significantly affected by the content of the prompt image. In terms of image reconstruction quality, the images generated by GenLV have clear textures and details. Conversely, Painter and PromptGIP may suffer from blurring or blocking artifacts, particularly for image restoration tasks. Overall, the above results show the superiority of GenLV in visual quality for dealing with various low-level tasks.

Table 1: Quantitative results on image restoration tasks. #: public released model. ★: trained with only restoration tasks. †: trained with all 30 low-level vision tasks. GN: Gaussian noise. PN: Poisson noise. ViT-VPIP: ViT backbone adopted in the VPIP framework. Our GenLV can also be represented as X-Restormer-VPIP. PSNR↑ (dB) is calculated as the quantitative metric.

| | GN | PN | S&P Noise | GB | JPEG | Ringing | R-L | Inpainting | SimpleRain | ComplexRain | Haze |
|---|---|---|---|---|---|---|---|---|---|---|---|
| Real-ESRGAN# | 25.38 | 26.57 | 21.50 | 21.49 | 25.21 | 24.64 | 21.71 | 14.06 | 16.10 | 21.01 | 11.86 |
| PromptIR★ | 28.86 | 31.48 | 36.45 | 24.56 | 26.77 | **27.85** | 31.31 | **28.11** | 30.76 | 24.08 | 16.85 |
| PromptGIP★ | 26.48 | 27.76 | 28.08 | 22.88 | 25.86 | 25.69 | 27.05 | 25.28 | 25.79 | 24.33 | 24.55 |
| ViT★ | 24.67 | 25.39 | 23.71 | 22.17 | 24.76 | 23.89 | 24.09 | 23.11 | 23.21 | 23.04 | 24.91 |
| ViT-VPIP★ | 26.14 | 27.20 | 25.43 | 24.13 | 26.19 | 25.98 | 26.98 | 25.03 | 25.51 | **24.79** | 24.06 |
| X-Restormer★ | 28.70 | 31.36 | 35.33 | 24.13 | 26.68 | 26.88 | 30.01 | 27.68 | 29.65 | 24.39 | 16.73 |
| GenLV★ (ours) | **28.99** | **31.69** | **36.63** | **24.58** | **26.91** | 27.74 | **31.50** | **28.11** | **31.10** | 24.71 | **28.91** |
| Painter† | 24.28 | 24.41 | 24.93 | 21.55 | 22.30 | 23.58 | 24.36 | 22.52 | 22.42 | 23.14 | 20.20 |
| PromptGIP† | 23.63 | 23.98 | 25.05 | 20.84 | 22.21 | 23.86 | 24.94 | 22.11 | 23.16 | 21.79 | 21.90 |
| ViT-VPIP† | 25.30 | 26.15 | 24.41 | 22.74 | 25.35 | 24.62 | 25.24 | 23.73 | 24.00 | 23.70 | 24.04 |
| GenLV† (ours) | **28.28** | **30.80** | **33.47** | **23.14** | **26.06** | **25.50** | **27.51** | **26.66** | **27.68** | **25.13** | **28.65** |

Table 2: Quantitative results on image enhancement and stylization tasks. PSNR↑ (dB) is calculated as the quantitative metric.

| | LowLight | LLF | Retouching | ICC | ICE | MTM | SDR2HDR | HDR2SDR | PencilDraw | Photographic | RTV |
|---|---|---|---|---|---|---|---|---|---|---|---|
| Painter† | 20.19 | 23.98 | 18.29 | 21.62 | 15.89 | 21.51 | 25.63 | 20.56 | 16.79 | 22.68 | 26.69 |
| PromptGIP† | 18.60 | 25.40 | 20.44 | 24.29 | 16.16 | 20.84 | 26.40 | 18.87 | 17.74 | 21.68 | 30.29 |
| ViT-VPIP† | 22.16 | 23.78 | 22.01 | 27.70 | 16.86 | 26.10 | 27.89 | 23.91 | 19.56 | 22.30 | 31.89 |
| GenLV† (ours) | **22.79** | **27.49** | **23.51** | **35.18** | **17.39** | **31.70** | **36.20** | **36.24** | **20.03** | **23.68** | **32.85** |

Table 3: Quantitative results on edge detection tasks. Mean absolute error↓ is calculated as the quantitative metric.

| | Canny | Laplacian | PED |
|---|---|---|---|
| Painter† | 31.36 | 7.06 | 9.55 |
| PromptGIP† | 19.48 | 4.06 | 9.36 |
| ViT-VPIP† | 27.68 | 5.49 | 8.44 |
| GenLV† (ours) | **8.30** | **1.28** | **7.23** |

Table 4: Standard deviation of the performance computed based on 20 different prompt images. PSNR (dB) is calculated as the quantitative metrics.

| | GN | GB | LowLight | ICC | PencilDraw | RTV |
|---|---|---|---|---|---|---|
| Painter† | 2.3930 | 1.8845 | 1.8865 | 1.9573 | 1.1820 | 2.6163 |
| PromptGIP† | 3.1035 | 2.2893 | 0.6766 | 0.6311 | 1.4200 | 1.3130 |
| GenLV† | **0.1033** | **0.0208** | **0.0399** | **0.0512** | 0.5518 | **0.0195** |

## 4.4 Exploration of Task Prompt

The above results have demonstrated the advantages of our prompt mechanism compared to existing methods from quantitative and qualitative perspectives. In this section, we conduct more experiments to further illustrate the effectiveness and explore the limitation of the task prompt in our method.

**Influence of Different Prompts.** To explore the influence on the quantitative performance for different prompt images, we randomly select 20 prompt image pairs for each task and calculate the performance on the corresponding test sets. Then, we compute the standard deviation of the 20 performance results for each task, as shown in Table 4. We can see that except for PencilDraw, the standard deviations are around or lower than 0.1dB. This shows that our method is stable in performance for different prompts.

**Task Prompt on Complex Situations.** We conduct further experiments to investigate the effectiveness of task prompt on complex situations. In Figure 6(a), we exhibit the outputs for images subjected to mixed degradation. The results show that the task prompt

successfully guide the mapping under this situation, and our method has the capability to deal with tasks with mixed degradation. In Figure 6(b), we present the results for cross-domain prompt. Utilizing Canny edge detection and LLE prompts, we instruct the model to process the noisy images. We can see that our model accurately execute the target task according to the visual prompts other than perform denoising. In Figure 6(c), we show the results on processing mixed degraded images using single-task prompts. The first row present the application of a denoising prompt to a low-light, noisy image. In the second row, we show that a deraining prompt is applied to a blurry image rain streaks. It can be see that the task-specific prompts effectively guide the model to perform the target task. All these results demonstrate the effectiveness of the visual task prompt in our method across a variety of complex situations.

**Mismatch Test.** We conduct mismatch test to illustrate the impact of the prompt on the model under special scenarios, as shown in Figure 7. The first row demonstrates providing the deblurring prompt to a clean image. In the second and third rows, we provide

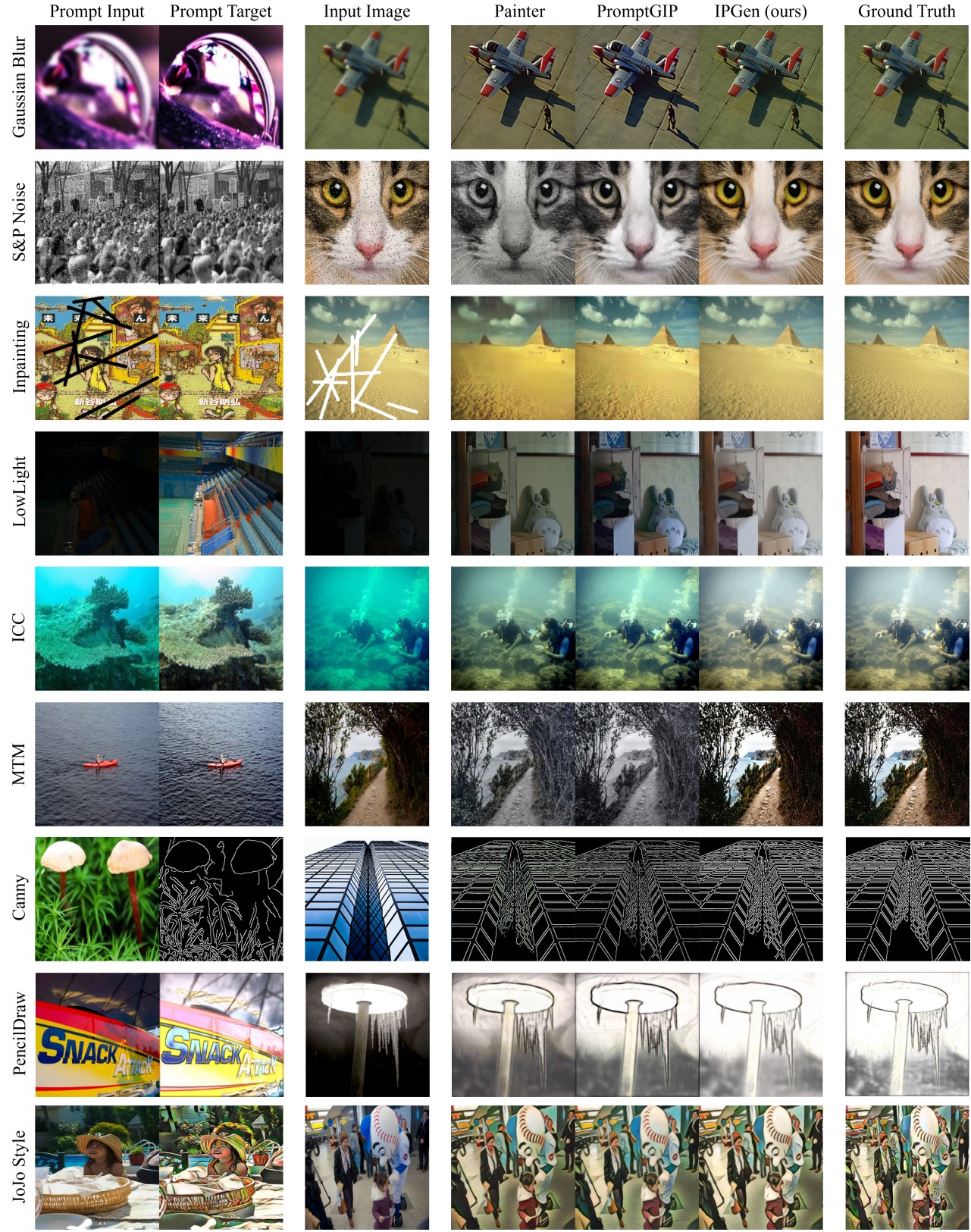

**Figure 5: Visual results of different models on various low-level vision tasks.**

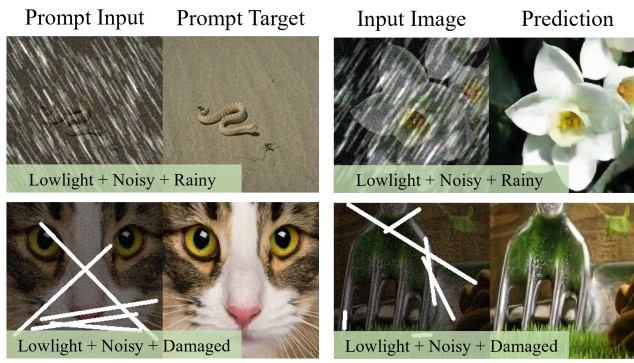

Prompt Input  Prompt Target   Input Image  Prediction

Lowlight + Noisy + Rainy  Lowlight + Noisy + Rainy

Lowlight + Noisy + Damaged  Lowlight + Noisy + Damaged

(a) Results for mixed degraded images.

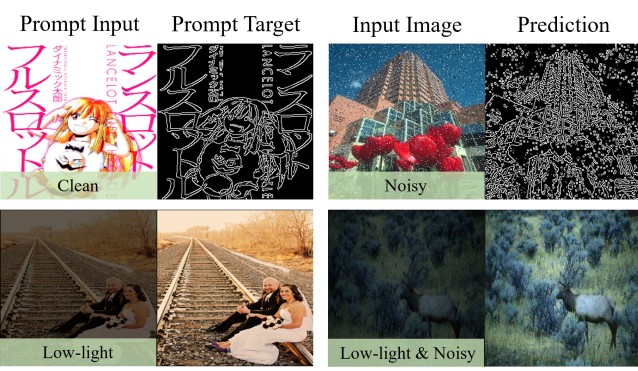

Prompt Input  Prompt Target   Input Image  Prediction

Clean  Noisy

Low-light  Low-light & Noisy

(b) Results for images based on cross-domain prompts.

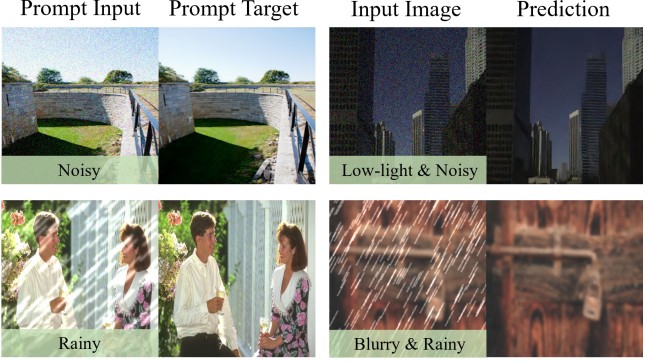

Prompt Input  Prompt Target   Input Image  Prediction

Noisy  Low-light & Noisy

Rainy  Blurry & Rainy

(c) Results for mixed degraded images on single-task prompts.

**Figure 6: Results for task prompts on complex situations.**

deJPEG and denoising prompts for low-light and blurry images, respectively. Ideally, we hope that the model do not execute the wrong prompt (this is reasonable from the perspective of the prompt cross-attention mechanism). It is observed that the model ideally preserves the original input images instead of performing degradation removal in these three instances. However, the mismatch test does not consistently yield ideal outcomes. In the fourth row, the model conducts deraining when provided with an inpainting prompt. From this perspective, this indicates that the model still inevitably overfit some data or mappings during training.

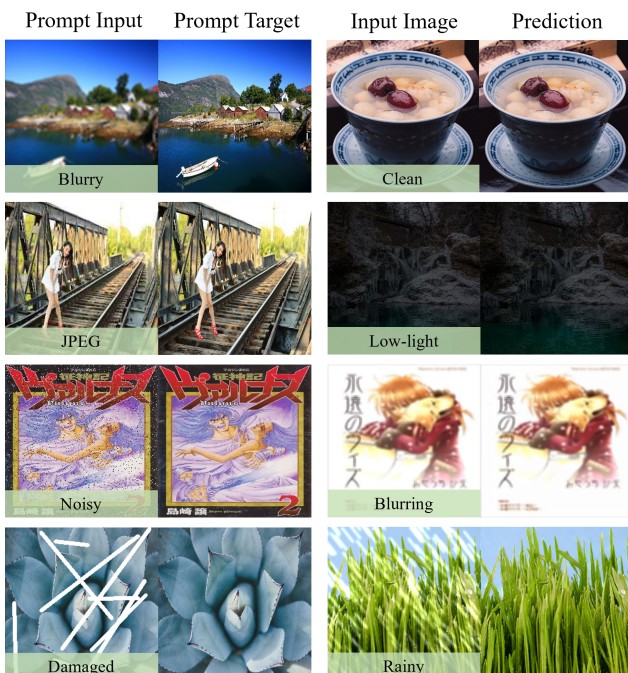

Prompt Input  Prompt Target   Input Image  Prediction

Blurry  Clean

JPEG  Low-light

Noisy  Blurring

Damaged  Rainy

**Figure 7: Results of the mismatch test.**

## 5 LIMITATIONS AND PROSPECTS

Our GenLV model demonstrates commendable performance in solving a broad range of low-level vision tasks, leveraging the visual prompt-based image processing framework and a powerful backbone network. Nonetheless, there are certain limitations and potential areas for further exploration that warrant attention. The working mechanism of this method is still to divide the task space via visual prompt, to achieve multi-task low-level vision. Despite the considerable improvement in performance compared to existing methods, we have to claim that the model currently still lacks the ability to generate satisfactory results for out-of-distribution unseen tasks. Recent study about large language models (LLMs) underscore that the effectiveness of LLMs largely depends on the quality, diversity, and quantity of the training data [43]. However, the task variety, model size (~30M), data scale (~140W) are not sufficient for GenLV. We hope that future studies involving larger models, broader tasks and data will yield more surprising results.

## 6 CONCLUSION

In this paper, we introduce a low-level vision generalist model, GenLV, which is capable of addressing various low-level vision tasks. Our approach involves the design of an image processing framework based on visual task prompt, VPIP, which enables the model to accommodate multiple tasks with different target domains. In addition, this framework allows the flexibility to incorporate a powerful backbone network that is suitable for low-level vision tasks, resulting in superior image reconstruction quality. Experimental results demonstrate that our GenLV can effectively manage 30 diverse low-level vision tasks and significantly outperform existing methods quantitatively and qualitatively.

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
