# OpenReview forum: "Learning A Low-Level Vision Generalist via Visual Task Prompt"
_acmmm.org/ACMMM/2024/Conference — MM2024 Poster_

### Official Review · Reviewer_gkVK · 2024-05-05

**Rating:** 4
**Confidence:** 3

**Summary:**

The paper presents GenLV, a versatile low-level vision model that excels at various image processing tasks through a Visual Task Prompt-based Image Processing (VPIP) framework. GenLV leverages a cross-attention mechanism and a powerful backbone network, X-Restormer, to process tasks with different input-target domains effectively. Trained on 30 diverse tasks, it outperforms existing methods in both quantitative and qualitative metrics, showcasing robustness across prompts. However, it has limitations with out-of-distribution tasks.

**Strengths:**

1. It adapts to various low-level vision tasks through a visual task prompt approach.
2. It shows improved performance over existing methods for the tasks it was trained on.
3. The prompt cross-attention mechanism offers a more efficient way to integrate task-specific information.

**Limitations:**

1.  It is crucial to conduct an ablation study that compares a single unified model, trained across all subtasks, with individual models trained for each specific subtask.  It is necessary to verify whether the unified model shows a decline in performance compared to the specialized model.

3.  The author's review of pertinent literature concerning task prompts appears to be inadequate, as works such as Emu edit, PowerPaint, and Diff-Plugin have been omitted.  It is essential to establish whether the task prompting methodology proposed in this paper holds any merits when compared to the aforementioned studies.

2.  Furthermore, it is recommended that the author provide an extensive presentation of image restoration results in real-world scenarios, accompanied by user studies.  It is worth noting that the domain of image restoration tends to have limited training and testing datasets.  Consequently, researchers should prioritize assessing the practical implications of their models in genuine situations.

**Suitability:**

2

---

### Official Review · Reviewer_hchM · 2024-05-25

**Rating:** 4
**Confidence:** 2

**Summary:**

The paper designs and trains a low-level vision model based on in-context learning, capable of performing 30 different low-level vision tasks.

**Strengths:**

The methodology and process of the paper are clearly described and well-organized.

The paper uses in-context learning to address low-level vision tasks. Compared to the latest similar method, such as PromptGIP, the proposed model covers a wider range of low-level tasks. And the author also provide experiment results on different backbones and task settings.

**Limitations:**

1. What datasets were used for training the network? How many images in total were used? Were these datasets used simultaneously for training? On which datasets were the individual tasks tested, and what were the quantities? What are the image sizes?

2. The paper presents results in terms of PSNR. Could the authors also provide some perceptual metrics, such as LPIPS, NIQE, BRISQUE, etc.?

3. I am curious whether the authors have analyzed the relationships between different tasks. For example, is there any mutual benefit between tasks, or are there conflicts between some tasks? For instance, the output of image feature extraction and image enhancement tasks can be quite different. Would learning these two tasks simultaneously be more beneficial than learning them separately?

**Suitability:**

2

---

### Official Review · Reviewer_jRyf · 2024-06-07

**Rating:** 4
**Confidence:** 3

**Summary:**

The paper presents a novel framework named Visual task Prompt-based Image Processing (VPIP) for addressing a variety of low-level vision tasks. The proposed model, GenLV, integrates visual prompts into the main image processing network, allowing the model to handle different input-target domain pairs flexibly. The authors employ a powerful backbone network, X-Restormer, to ensure high-quality image reconstruction and introduce a cross-attention mechanism to facilitate information interaction between the input and the prompts. Experimental results on 30 diverse tasks demonstrate the superiority of GenLV over existing methods, both quantitatively and qualitatively.

**Strengths:**

1.Innovative Framework: The VPIP framework is a significant contribution, providing a flexible solution to handle various low-level vision tasks using visual prompts.
2.High Performance: The proposed GenLV model shows substantial improvement over existing methods in both image restoration and enhancement tasks.
3.Comprehensive Evaluation: The authors conduct extensive experiments on a wide range of tasks, including image restoration, enhancement, edge detection, and stylization, showcasing the versatility and robustness of GenLV.
4.Detailed Analysis: The paper provides thorough quantitative and qualitative analyses, including ablation studies and visual comparisons, to support the effectiveness of the proposed approach.
5.Efficient Attention Mechanism: The introduction of the prompt cross-attention mechanism reduces computational cost and enhances the model's capability to process diverse tasks.

**Limitations:**

1.Dependency on Task-Specific Prompts: The model's performance heavily relies on the quality and relevance of the visual prompts, which may limit its generalization to unseen tasks or domains.
2.Training Complexity: Training the GenLV model on 30 diverse tasks requires significant computational resources and careful tuning of hyperparameters, which might be a barrier for broader adoption.
3.Model Size and Data Requirements: The current model size and the scale of training data might not be sufficient for achieving even higher performance, suggesting the need for further research with larger models and datasets.
4.Detailed description of Method: The description integration of Prompt representation and Low-level Vision Generalist Model is not detailed and clear enough.

**Suitability:**

2

---

### Official Review · Reviewer_B79v · 2024-06-07

**Rating:** 5
**Confidence:** 3

**Summary:**

This paper introduces the Visual task Prompt-based Image Processing (VPIP) framework to address challenges in handling diverse low-level vision tasks. The generalist model GenLV, built on the VPIP framework and trained on 30 different tasks, shows significant improvements over existing approaches.

**Strengths:**

1. The motivation for this work is reasonable, aiming to create a unified model framework for low-level tasks.

2. The paper is well-written, logically clear, and easy to understand.

3. Experiments show that the proposed model performs well.

**Limitations:**

1. It is hoped that the authors can provide further introduction on the selection and definition of low-level tasks.

2. The authors claim that the proposed prompt cross-attention method can improve experimental efficiency; this requires additional comparative data for support.

3. Apart from image enhancement tasks, the comparison with specific task methods is lacking for several other tasks.

4. It would be beneficial to provide further details on the specific content of the task prompts.

**Suitability:**

3

---

### Official Review · Reviewer_RZon · 2024-06-09

**Rating:** 5
**Confidence:** 3

**Summary:**

This paper proposes a low-level vision generalist model GenLV, to employ the visual task prompt to process tasks with different input-target domains. A prompt cross-attention mechanism is introduced to deal with the information interaction between the input and prompt information.

**Strengths:**

1. The trained low-level vision generalist model GenLV, can successfully process the various tasks with different input and target domains.
2. The quantitative and qualitative comparisons demonstrate the superiority of the proposed framework.

**Limitations:**

1. What is the inference time, compared with previous methods? If it is too large than previous methods, it may not be a fair comparison.

2. Any failure cases to show and discuss?

**Suitability:**

3

---

### Meta-Review · Program_Chairs · 2024-07-12

**Recommendation:** Accept (Poster)
**Confidence:** 5

**Metareview:**

This paper has been thoroughly evaluated by 5 reviewers. They have all raised many questions in their comments, which have been nicely addressed by authors in the rebuttal -- it caused one score to raise. Everyone appreciates the originality of the approach, the quality of the writing, the extensiveness of the experiments. The contribution has a series of limitations, as highlighted in the reviews, but enough merits to pass the bar.

For these reasons, the recommendation is ACCEPT, poster.